# THE BENEFITS OF INCORPORATING SHAPE PRIORS IN CONTRASTIVE LEARNING

**Junru Zhao** *
Department of Electrical and Computer Engineering
Carnegie Mellon University
Pittsburgh, PA 15213, USA
`junruz@andrew.cmu.edu`

**Tianqin Li** *
Department of Computer Science
Carnegie Mellon University
Pittsburgh, PA 15213, USA
`tianqinl@cs.cmu.edu`

**Tai-Sing Lee**
Department of Computer Science
Carnegie Mellon University
Pittsburgh, PA 15213, USA
`tai@cs.cmu.edu`

## ABSTRACT

Human babies develop the ability of figure-ground segregation based on motion, luminance and color cues early on during infancy. The availability of the global form or shape of the objects is known to facilitate rapid learning of lexical categories in babies. Here, we explored the use of shape prototypes, computed by momentum clustering the global forms of objects, to bootstrap a form of self-supervised learning, called contrastive learning, to mimic human learning. We found that shape prototypes can play a positive role in speeding up representation learning by highlighting the importance of object boundaries and forms at the initial learning phase but might hinder learning detailed features for object recognition. Thus, a hybrid of "coarse-to-fine" or "shape-to-texture" training regimes that foster learning global shapes and local features produces high-performance object recognition systems with global shape sensitivity.

## 1 INTRODUCTION

The global shapes of objects play a pivotal role in human object recognition and representational learning. Many studies have revealed that human object recognition uses shape as a principal discriminative cue with few costs (Wagemans et al. (2008); Elder & Velisavljević (2009)). Children, for example, have been found to learn new lexical categories much faster when the categories are organized by shapes (Gershkoff-Stowe & Smith (2004)). How can global shape representation be learned? How can global shape representations facilitate the development of categorical recognition? Neural networks learned to perform object recognition have been found to rely heavily on local features and texture cues rather than global shapes of objects (Islam et al. (2021); Geirhos et al. (2022; 2020)). Currently, there is active research in the deep learning community on the strategies and training paradigms that will endow deep networks with stronger "shape bias" (Geirhos et al. (2020; 2022); Wen et al. (2023)). The presumption is that having shape representations that are aligned with human perception will make deep networks more flexible and generalizable, resulting in better object recognition performance. Most of those studies, however, focus on the supervised learning and the progress of enhancing shape bias through self-supervised learning is limited.

In this paper, we investigated the impact of incorporating global shape priors in contrastive representational learning in deep neural networks. We introduced a novel contrastive learning framework called **Shape Prototype Contrastive Learning (S-PCL )**. In this framework, global shape in the form of the silhouette of an object, extracted using existing methods on figure-ground segregation, is used as to augment the input training image in contrastive learning. Shape prototypes, the cluster

---

*Authors Junru Zhao and Tianqin Li contributed equally to this work.

centroids of silhouette shapes of objects in the data set in the embedding space, are learned with momentum clustering and then used to organize the embedding space of the original input images. We found that deep networks learned with these shape prototype priors exhibit stronger shape representations that are more aligned with human perception. Furthermore, we found that S-PCL accelerates the learning process, particularly in the early stage of development, reminiscent of the impact of global shapes in children's lexical categorical learning. This also alleviate the problem that most of current contrastive learning frameworks require a long training period to achieve a a satisfying representation space (He et al. (2020); Grill et al. (2020); Li et al. (2021)).

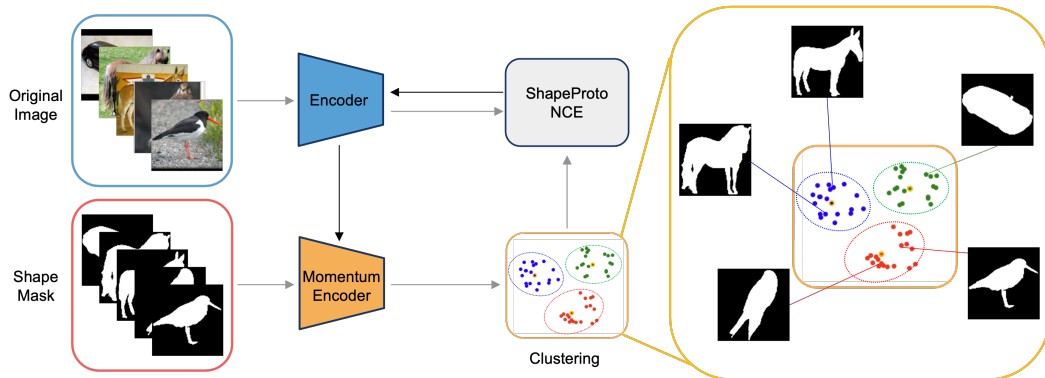

Figure 1: Training framework of Shape Prototype Contrastive Learning. The original images are input to encoder and the shape silhouettes are input to the momentum encoder. The clustering is performed on the embedding of shape silhouettes. The gray arrows represent feed-forward and the black arrows represent back-propagation.

## 2   RELATED WORK

Our work is closely related to two main topics: contrastive learning and shape bias of CNN.

**Contrastive learning**   The contrastive learning is a training framework of self-supervised learning where a model is trained to distinguish between similar and dissimilar embedding pairs of instances . the instance-wise contrastive learning (van den Oord et al. (2019); Wu et al. (2018); He et al. (2020); Grill et al. (2020); Chen et al. (2020)) aims to learn a embedding space where instances with similarities are pulled closer while dissimilar instances are pushed far away. This goal is always accomplished by optimizing the contrastive loss function. Another type of contrastive learning is combined with clustering (Caron et al. (2021); Li et al. (2021)). In this kind of framework, the concept of prototype is introduced to guide the formation process of the embedding space. The prototype contrastive learning framework produces more separate clusters in the embedding space.

**Shape bias**   Shape bias is one significant difference between CNN-based models and humans where CNNs has texture bias while human has shape bias (Geirhos et al. (2022; 2021)). In supervised learning, many studies show that enhancing shape bias of CNN through shape-related data augmentation (Wen et al. (2023)) can result in improvement of prediction robustness and domain generalization (Geirhos et al. (2022); Li et al. (2023)). For self-supervised learning, although the existence of texture bias of CNN is also identified (Geirhos et al. (2020)), the approaches and benefits of enhancing shape bias in self-supervised learning is still unclear, which motivates our work.

## 3   APPROACH

### 3.1   PRELIMINARIES

**Instance-wise contrastive learning**   In instance-wise contrastive learning, e.g. MoCoV2, given training images $X = \{x_1, x_2, ..., x_n\}$ and an encoder $f(\cdot)$ encoding $X$ to embedding $V =$

$\{v_1, v_2, ..., v_n\}$, i.e. $f(x) = v$, the optimal encoder $f$ is found by optimizing the InfoNCE (van den Oord et al. (2019)) loss function:

$$\mathcal{L}_{\text{InfoNCE}} = \sum_{i=1}^{n} -\log \frac{\exp(v_i \cdot v_i')/\tau)}{\sum_{j=0}^{r} \exp(v_i \cdot v_j'/\tau)} \tag{1}$$

where $v_i'$ is one positive sample of $v_i$, $v_j', j \neq i$ are negative samples of $v_i$, and $\tau$ is the temperature. In MoCoV2, positive and negative samples $v'$ are obtained by inputting train data to the momentum encoder $f'(\cdot)$, i.e. $f'(x) = v'$.

**Prototype contrastive learning**  In PCL, the embedding of training images obtained by momentum encoder are further clustered. The centroids of clusters are called prototypes. The optimal $f(\cdot)$ is found by optimizing the ProtoNCE (Li et al. (2021)) loss function:

$$\mathcal{L}_{\text{ProtoNCE}} = \sum_{i=1}^{n} -\log \frac{\exp(v_i \cdot c_p)/\phi_p)}{\sum_{j=0}^{r} \exp(v_i \cdot c_j/\phi_j)} \tag{2}$$

where $c_p$ is the prototype of the cluster $p$ that $v_i$ belongs to, $c_j, j \neq p$ are centroids of other clusters, and $\phi_j$ is the concentration estimation of cluster $j$. The clustering is performed for multiple times with different number of clusters, therefore the loss for optimization is the mean of multiple runs.

## 3.2 SHAPE PROTOTYPE CONTRASTIVE LEARNING

This section introduces our proposed contrastive learning framework, Shape Prototype Contrastive Learning (S-PCL), designed to incorporate shape silhouette of the object in the training phase so as to explicitly enhance the shape bias of the embedding space.

**Preprocessed shape silhouette**  In S-PCL, we choose to use the silhouette to represent the object shape which segregates the salient object and background, as shown in Figure.1. The silhouettes are generated offline as a part of data preprocessing. Studies show that human can instantly capture the shape of an object at minimal cost (Elder & Velisavljević (2009)) and figure-ground representation is readily available to the human visual system (Lamme (1995); Zipser et al. (1996); Lee et al. (1998)). This provides the ground for preprocessing to obtain the shape silhouette and input it directly into the network.

**Shape Prototype**  The term "shape prototype" refers to the representative embedding for a group of shape silhouettes. To find the shape prototype, we perform K-Means clustering. The set of shape silhouette masks, $M = \{m_1, m_2, ..., m_n\}$, is encoded by the momentum encoder to the shape embedding set, $U = \{u_1, u_2, ..., u_n\}$, i.e. $u = f'(m)$. K-Means clustering is employed on the set $U$, producing $K$ clusters and a set of centroids, $S = \{s_1, s_2, ..., s_K\}$. The centroid of the cluster is representative for all shape embeddings in the cluster, thus is considered as the shape prototype. Subsequently, the shape bias is introduced through optimization using ShapeProtoNCE, which contrasts shape prototypes and the original image embeddings.

**ShapeProtoNCE**  Given the set of shape embedding $U$, the set of shape prototypes $S$, and the set of embeddings of original training images $V$, the ShapeProtoNCE is defined:

$$\mathcal{L}_{\text{ShapeProtoNCE}} = \sum_{i=1}^{n} -\log \frac{\exp(v_i \cdot s_p)/\phi_p)}{\sum_{j=0}^{r} \exp(v_i \cdot s_j/\phi_j)} \tag{3}$$

where $s_p$ is the prototype of the cluster $p$ to which $u_i$ belongs, $s_j, j \neq p$ are shape prototypes representing other shape clusters, and $\phi_j$ is the concentration estimation of cluster $j$. Similar to the previous work (Li et al. (2021)), we perform clustering for $N$ times with different $K_i, i = 1, 2, ..., N$ and average the losses. Also, we find that combining InfoNCE and ShapeProtoNCE is beneficial. Therefore, the loss function we use is:

$$\mathcal{L} = \mathcal{L}_{\text{InfoNCE}} + \frac{1}{N} \sum_{i=1}^{N} \mathcal{L}_{\text{ShapeProtoNCE}, K_i} \tag{4}$$

**S-PCL+X hybrid approach**   Taking the inspiration from the fact that humans learn new nouns based on the shape of objects during the infancy, and noticing S-PCL significantly speeds up the learning process but limiting the network later on, we decided to use S-PCL in the initial training stage to organize the embedding space based on global shape priors and then switch to the other state-of-the art instance-wise and prototype contrastive learning framework for further fine-tuning in the later stage. We find this sequential hybrid training strategy of "coarse-to-fine" and "shape-to-texture" to be beneficial. We denote the S-PCL warmed-up method as S-PCL+X, for example, MoCoV2 warmed-up by S-PCL is denoted by S-PCL+MoCoV2.

### 3.3   Implementation details

The experiments are conducted on three different datasets: Imagenette, ImageNet-100[1], the subset of ImageNet-1K (Deng et al. (2009)), and STL10 dataset (Coates et al. (2011)). The commonly-used instance-wise MoCoV2 and PCL are served as baselines.

The silhouette mask of the object is generated offline using TRACER (Lee et al. (2022)). The ResNet18 (He et al. (2016)) is adopted as the encoder and the output dimension of the last fully-connected layer is 256-D. We follow the data augmentation methods in previous works (He et al. (2020); Li et al. (2021)). We train the ResNet18 for a set of numbers of epochs $\{50, 100, 150, 200, 300, 400\}$. We use SGD as the optimizer, with a weight decay of 1e-4, a momentum of 0.9, and a batch size of 128. The initial learning rate is 0.3 for ImageNet-100 and STL10, 0.03 for imagenette, and they all follow the cosine learning rate schedule.

In S-PCL training, the model is warmed-up in the first 20 epochs in the same way as stated in (Li et al. (2021)). We set $\tau = 0.1, \alpha = 10$, and the number of clusters $K = \{2000, 4000, 6000\}$. In order to have fair comparison, we use the same hyperparameters as S-PCL for the PCL training. MoCoV2 uses the same $\tau = 0.1, \alpha = 10$. For all three frameworks, we set the number of negative samples $r = 1024$ when training on ImageNet-100 and STL10, and $r = 128$ when training on imagenette. When S-PCL serving as a warm-up approach, the first 100 epochs is trained by S-PCL framework. Then it will be trained using PCL or MoCoV2 following the same learning rate schedule in the subsequent epochs. Hyper-parameters are the same as above.

## 4   Results

### 4.1   Linear classification

We evaluate the learned representation on image classification tasks. During the inference phase, the shape silhouette is bot involved in order to ensure a fair comparison with other contrastive frameworks. A linear classifier is trained based on the frozen image representations using labeled data. For Imagenette and ImageNet-100, entire labeled training set is used and accuracy on the validation set is reported. For STL10, we use labeled training set and test on the test set. Detailed numerical results can be found in appendices.

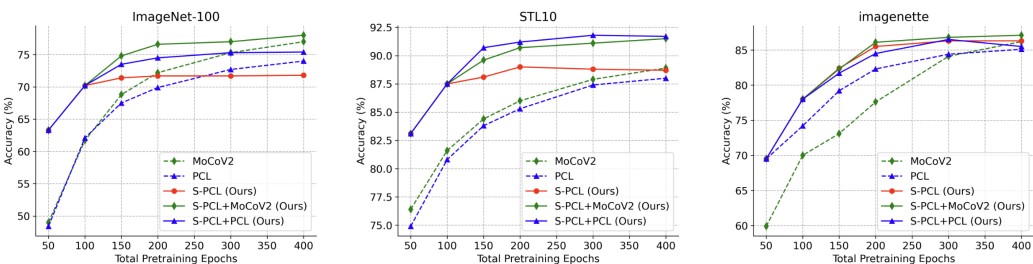

Figure 2: Linear classification accuracy v.s. total pretraining epochs. Since the S-PCL+MoCo and the S-PCL+PCL are warmed-up by S-PCL in the first 100 epochs, therefore their curves overlap.

---

[1]The dataset used here is reorganized by adjusting the train-validation split to a 7:3 ratio to intensify the challenge

## 4.2 INCORPORATING SHAPE INFORMATION ACCELERATES LEARNING

From Figure.2, it can be observed that the MoCoV2 and the PCL achieve a high accuracy, outperforming S-PCL on ImageNet-100 after 400 epochs of training, however the improvement rate is slow during the training. In contrast, S-PCL attains a relatively high accuracy within the first 100 epochs but plateaus and even performs worse than others in the later stage, suggesting that the shape, as a coarse visual cue, is beneficial in the early training while it is the finer cue such as local textures that pushes the performance to the next level in the later training. We can observe the similar trend that S-PCL based models converge faster on STL10 and imagenette as well.

Importantly, the results of S-PCL+MoCoV2 and S-PCL+PCL, trained using the "shape-to-texture" sequential regime, are encouraging. They achieve high accuracy levels comparable to MoCoV2 and PCL but with only half the training epochs, indicating that shape priors accelerate early learning. Notably, after 400 epochs of training, our proposed regime ends up having the highest accuracy. Overall, our experimental results demonstrate that learning shape priors in the early stages can effectively bootstrap the training of other contrastive learning frameworks, such as MoCoV2 and PCL.

## 4.3 VALIDATING SHAPE BIAS

We find it necessary to confirm our approach introduces shape bias to the model and it is the shape bias that leads to such a result. To this end, we quantitatively evaluate linear classifiers on style-transferred images and present qualitative evidence in the form of sensitivity maps computed by SmoothGrad (Smilkov et al. (2017)).

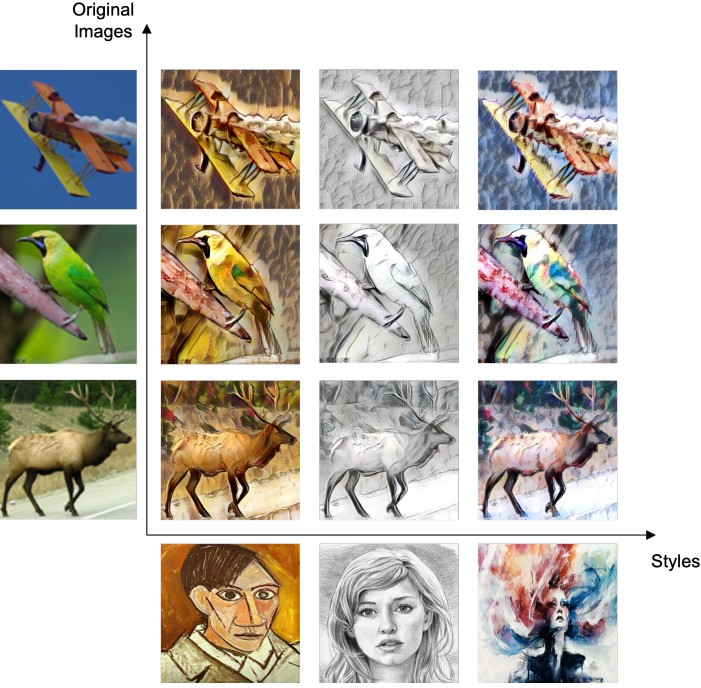

Figure 3: Illustrations of style-transferred images.

### 4.3.1 TEST ON STYLE-TRANSFERRED IMAGES

The out-of-domain robustness is always associated with the shape bias (Geirhos et al. (2022; 2020)). The style-transferred image is an out-of-domain representation of the original image and is considered to preserve the shape of object but have distinct texture and style Based on the same intuition, we evaluate the classification accuracy of the linear classifier trained on style-transferred test sets. The ImageNet-100 validation set and STL10 test set are transferred to three different styles, as shown

in Figure.3. Table.1 and 2 report results, note that results of the style-transferred set are the mean of 3 different styles. On both datasets, the models trained or warmed-up by S-PCL end up having higher top-1 and top-5 accuracies on style-transferred validation even though the accuracies on the original sets are much lower, indicating that S-PCL produces models that are more robust against the drastic style and texture changes, i.e. exhibiting stronger shape bias.

Table 1: Evaluation on the style-transferred ImageNet-100 validation set. The highest values in each column are in bold.

| | EPOCHS PRETRAINED | ORIGINAL | | STYLE | |
|---|---|---|---|---|---|
| | | TOP-1 ACC (%) | TOP-5 ACC (%) | TOP-1 ACC (%) | TOP-5 ACC (%) |
| S-PCL | 400 | 71.8 | 90.9 | **30.4** | **53.4** |
| MOCOV2 | 400 | **77.0** | **93.8** | 26.0 | 47.1 |
| S-PCL+MOCOV2 | 200 | 76.4 | 93.1 | 28.8 | 51.2 |
| PCL | 400 | 74.1 | 91.9 | 26.8 | 48.8 |
| S-PCL+PCL | 200 | 74.5 | 92.1 | 30.0 | 52.1 |

Table 2: Evaluation on the style-transferred STL10 test set. The highest values in each column are in bold.

| | EPOCHS PRETRAINED | ORIGINAL | | STYLE | |
|---|---|---|---|---|---|
| | | TOP-1 ACC (%) | TOP-5 ACC (%) | TOP-1 ACC (%) | TOP-5 ACC (%) |
| S-PCL | 400 | 88.7 | 99.5 | **47.7** | **84.6** |
| MOCOV2 | 400 | 88.9 | **99.7** | 30.1 | 67.0 |
| S-PCL+MOCOV2 | 200 | 90.6 | **99.7** | 37.4 | 74.7 |
| PCL | 400 | 88.0 | 99.4 | 33.5 | 74.3 |
| S-PCL+PCL | 200 | **91.2** | **99.7** | 47.2 | 78.3 |

### 4.3.2 SENSITIVITY MAP

The sensitivity map reveals the image regions that contribute significantly to the prediction. We apply SmoothGrad to the linear classifier learned in the inference phase by each contrastive learning framework for 400 epochs on ImageNet-100 dataset and present the resulting sensitivity maps.

**Sensitivity on original images** First, we evaluate sensitivity maps on original images, as depicted in Figure.4. Example (a) features a spider with a complex shape as well as rich textures, while the other three examples showcase objects with relatively simple shapes but rich textures. Specifically, for example (a), only S-PCL appears sensitive to the entire body of the spider, indicating that shape plays a crucial role in the predictions made by ResNet18 pretrained with S-PCL. In the case of the other three examples, MoCoV2 and PCL exhibit a strong preference for specific textures, such as folds on the leather bag in (b), decorative patterns on the coffee cup in (c), and facial features of the animal in (d). In contrast, the sensitivity maps of S-PCL reveal distinct object silhouette boundaries, suggesting much greater attention to the contour of the object rather than its content. This implies that ResNet18 pretrained with S-PCL makes predictions primarily based on object shape, while MoCoV2 and PCL favor local features.

**Sensitivity on style-transferred images** In addition to visualizing sensitivity maps on original images, we also apply SmoothGrad to style-transferred images to further evaluate the shape bias. The style-transferred images preserve the shape while exhibiting different textures. It's important to note that style transferring not only alters the object's texture but also introduces new textures in the background, leading to intense responses in the background within the sensitivity maps of MoCoV2 and PCL, as shown in Figure.5. This further emphasizes that MoCoV2 and PCL exhibit a very

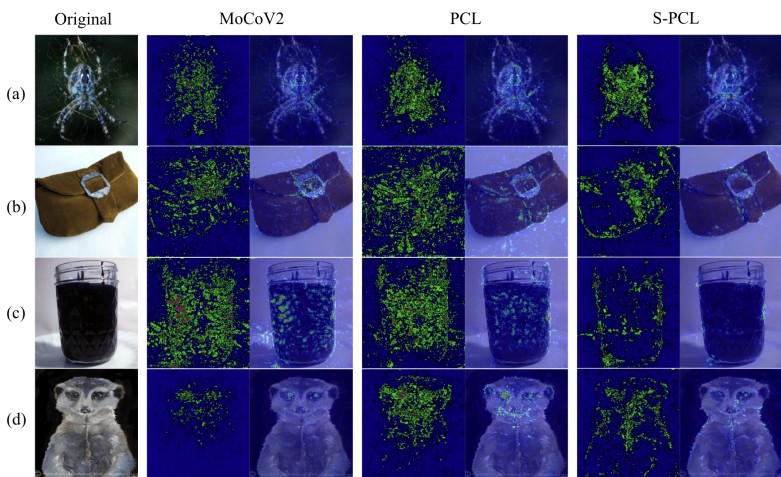

Figure 4: Sensitivity maps of ResNet18s pretrained by different self-supervised learning frameworks. For each framework, pure sensitivity map (left) and image with sensitivity map (right) are shown.

strong texture bias. On the other hand, S-PCL demonstrates more consistent sensitivity maps, still focusing on the coffee cup silhouette boundary and displaying greater robustness to texture noise in the background.

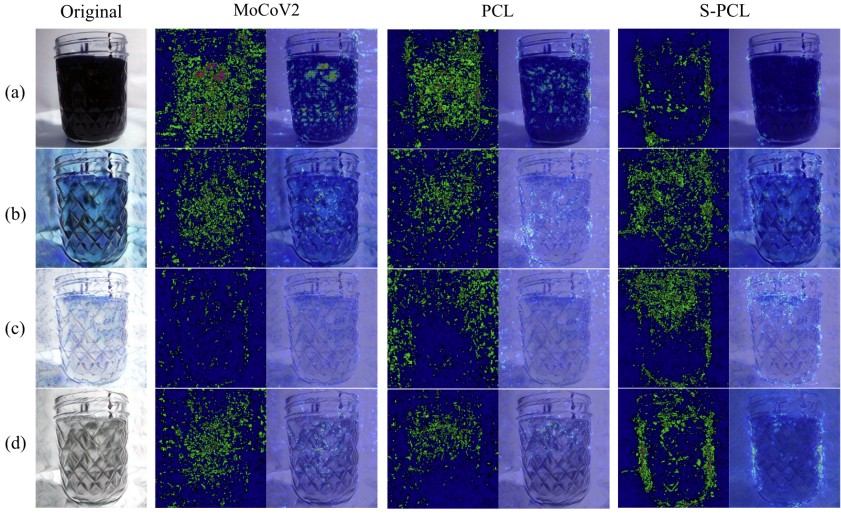

Figure 5: Sensitivity maps of ResNet18s pretrained by different self-supervised learning frameworks on images of the same content rendered in different styles.

## 5 DISCUSSION

In this paper, we proposed a new contrastive learning framework called Shape Prototype Contrastive Learning. Shape prototypes, learned across images over time, provided a "memory mechanism" to influence the embedding space in contrastive learning. During the learning phase, shape silhouettes of objects are provided with the training images as augmented input to the systems. During the inference phase, the shape silhouettes are not provided. This ensures that the proposed framework can be compared on equal ground against other contrastive learning paradigms in object recognition

performance. We found that having global shape prototype priors during training greatly accelerates the learning process in the early stage but hinders learning in the later stage (Figure.2). This suggests that the stereotypical shapes of objects can help guide the deep networks quickly into a better embedding space for distinguishing semantic categories more separable, but then later on, might actually hinder the network from learning the finer and subtler discriminating features useful for better recognition. Hence, we proposed a hybrid approach, using the shape prototype priors at the beginning but relinquishing their influence in the later stage. Using S-PCL as a warm up training procedure can effectively bootstrap the training of other contrastive learning frameworks, such as MoCoV2 and PCL. Our finding on the enhancement of categorical learning by auxiliary input of global shapes is reminiscent of the facilitation of global shapes in infant learning of lexical categories. Since self-supervised contrastive learning has been considered relevant for understanding human visual learning, our framework might thus share some underlying mechanisms on the role of global shape perception in representation learning.

We found our framework of providing the global shape input and the modulation learning the embedding space with remembered shape prototypes increase the global shape sensitivity of the deep networks, as evident in the increased "shape bias" or robustness to style transfer in object recognition as shown in Table.1 and 2. Feature attribution sensitivity analysis, as shown in Figure.4 and Figure.5, revealed that the S-PCL-trained networks indeed pay more attention to the contours of objects, even during inference when the silhouette is not provided as input to the network. This observation suggests that the network has learned to pay more attention to the global shapes or the boundaries of the objects because of this training with shape prototypes.

Figure-ground segregation is known to be computed in one of the earlier stages in the visual system, and figure-ground representation resulting from motion, luminance, color, and stereo are readily available to the visual systems even for infants. Therefore, figure-ground representation can be and probably should be used in representational learning for object recognition. In this paper, for evaluating the benefits of S-PCL training, we did not use figure-ground representation during inference. However, figure-ground representation is likely used together with other signals in object recognition in animals. Thus, it would be interesting in future study to investigate whether recognition performance can be enhanced when figure-ground is provided as auxiliary input during inference as well. It is also important to investigate the nature of the shape prototypes that have been learned and how and when they can continue to help object recognition. To better relate S-PCL to compositional learning, it will be useful to investigate the impact of having shape clustering at multiple layers of the networks rather than just at the highest level.

## 6  ACKNOWLEDGEMENT

This work was supported by an NSF grant CISE RI 1816568, NIH R01 EY030226-01A1 and a graduate student fellowship from CMU Computer Science Department. We also thank NSF AC-CESS Allocations (Project Number CIS230221) for computational resource support and Ziqi Wen for helpful discussion.

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

## A    FULL RESULTS OF LINEAR EVALUATION ON THREE DATASETS

Table.3-5 report the accuracies of linear evaluation of different models on different datasets. The highest one is in bold.

Table 3: Linear classification top-1 accuracy on ImageNet-100. At epochs 50 and 100, both S-PCL+MoCoV2 and S-PCL+PCL achieve the same accuracy as S-PCL, since the first 100 epochs serve as a warm-up phase for S-PCL.

| # OF EPOCHS | 50 | 100 | 150 | 200 | 300 | 400 |
|---|---|---|---|---|---|---|
| S-PCL | 63.3 | 70.2 | 71.4 | 71.7 | 71.7 | 71.8 |
| MoCoV2 | 49.0 | 61.7 | 68.8 | 72.2 | 75.3 | 77.0 |
| S-PCL+MoCoV2 | 63.3 | 70.2 | 74.8 | 76.6 | 77.0 | **78.0** |
| PCL | 48.4 | 62.1 | 67.5 | 69.9 | 72.7 | 74.0 |
| S-PCL+PCL | 63.3 | 70.2 | 73.5 | 74.5 | 75.3 | 75.4 |

Table 4: Linear classification top-1 accuracy on STL10. At epochs 50 and 100, both S-PCL+MoCoV2 and S-PCL+PCL achieve the same accuracy as S-PCL, since the first 100 epochs serve as a warm-up phase for S-PCL.

| # OF EPOCHS | 50 | 100 | 150 | 200 | 300 | 400 |
|---|---|---|---|---|---|---|
| S-PCL | 83.1 | 87.5 | 88.1 | 89.0 | 88.8 | 88.7 |
| MoCoV2 | 76.4 | 81.6 | 84.4 | 86.0 | 87.9 | 88.9 |
| S-PCL+MoCoV2 | 83.1 | 87.5 | 89.6 | 90.7 | 91.1 | 91.5 |
| PCL | 74.9 | 80.8 | 83.8 | 85.3 | 87.4 | 88.0 |
| S-PCL+PCL | 83.1 | 87.5 | 90.7 | 91.2 | **91.8** | 91.7 |

## B    ABLATION ON SHAPEPROTONCE

We carry out ablation studies on different combinations of loss functions on the STL10 dataset. Results are reported in Table.7. We find that using only ShapeProtoNCE results in lower accuracy on the original test set than combining InfoNCE and ShapeProtoNCE. However, it yields much higher accuracy on the style-transferred test set. This further suggests that using ShapeProtoNCE makes the model less prone to texture changes, while using InfoNCE makes the model prone to texture changes.

Table 5: Linear classification top-1 accuracy on Imagenette. At epochs 50 and 100, both S-PCL+MoCoV2 and S-PCL+PCL achieve the same accuracy as S-PCL, since the first 100 epochs serve as a warm-up phase for S-PCL.

| # OF EPOCHS | 50 | 100 | 150 | 200 | 300 | 400 |
|---|---|---|---|---|---|---|
| S-PCL | 69.5 | 78.0 | 82.4 | 85.5 | 86.3 | 86.3 |
| MoCoV2 | 59.9 | 70.0 | 73.1 | 77.6 | 84.1 | 86.2 |
| S-PCL+MoCoV2 | 69.5 | 78.0 | 82.3 | 86.1 | 86.8 | **87.1** |
| PCL | 69.5 | 74.2 | 79.2 | 82.3 | 84.4 | 85.1 |
| S-PCL+PCL | 69.5 | 78.0 | 81.7 | 84.5 | 86.5 | 85.5 |

Table 6: Results of ResNet18 trained by different combinations of loss functions.

| | ORIGINAL | | STYLE | |
|---|---|---|---|---|
| | TOP-1 ACC (%) | TOP-5 ACC (%) | TOP-1 ACC (%) | TOP-5 ACC (%) |
| SHAPEPROTONCE | 79.4 | 99.0 | **53.4** | **92.4** |
| INFONCE+SHAPEPROTONCE | **87.5** | **99.6** | 46.6 | 86.7 |

## C ABLATION ON TRAINING ORDER

We also explore the effect of the order of different training frameworks. The experiments are conducted on STL10 dataset. We find that using S-PCL first ends up with higher accuracy on the original data. Results on style-transferred data suggest that incorporating shape in the train does increase the shape bias regardless of the order yet we cannot tell which order is better from the perspective of improving shape bias given the current result.

Table 7: Results of ResNet18 trained by different orders of methods.

| | EPOCHS PRETRAINED | ORIGINAL | | STYLE | |
|---|---|---|---|---|---|
| | | TOP-1 ACC (%) | TOP-5 ACC (%) | TOP-1 ACC (%) | TOP-5 ACC (%) |
| MoCoV2 | 400 | 88.9 | **99.7** | 30.1 | 67.0 |
| MoCoV2+S-PCL | 100+100 | 87.8 | 99.5 | **43.2** | **86.1** |
| S-PCL+MoCoV2 | 100+100 | **90.7** | **99.7** | 37.4 | 74.7 |
| PCL | 400 | 88.0 | 99.5 | 33.5 | 74.3 |
| PCL+S-PCL | 100+100 | 88.0 | **99.7** | 42.1 | **83.8** |
| S-PCL+PCL | 100+100 | **91.2** | **99.7** | **47.2** | 78.3 |

## D MUTUAL INFORMATION BETWEEN THE CLUSTERING RESULT AND THE LABEL

We evaluate the clusters produced by PCL and S-PCL+PCL by calculating the adjusted mutual information (AMI) (Vinh et al. (2010); Li et al. (2021)) between the clustering assignments and ground-truth labels for ImageNet-100. As the Shape-PCL performs clustering on the shape mask and the input data of clustering is not consistent with PCL, so the AMI of Shape-PCL is not comparable with that of PCL. Comparing the AMI curves of PCL and S-PCL+PCL, we can also conclude that incorporating shape can accelerate learning. Interestingly, we observe the AMI of Shape-PCL would drop when trained for longer timer, this is probably the key to understand why Shape-PCL plateaus and worth studying in the future work.

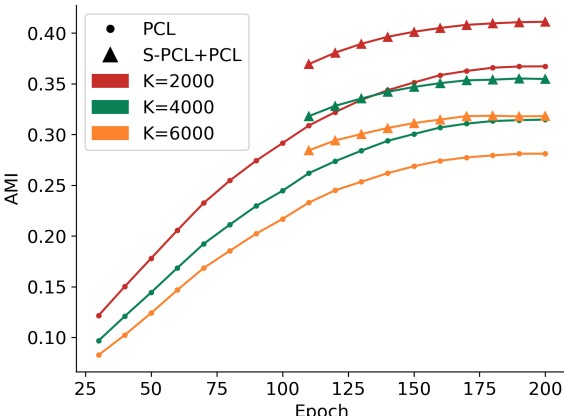

Figure 6: Adjusted mutual information between the clusters generated by PCL, S-PCL+PCL after 100 epoch and ground-truth labels.

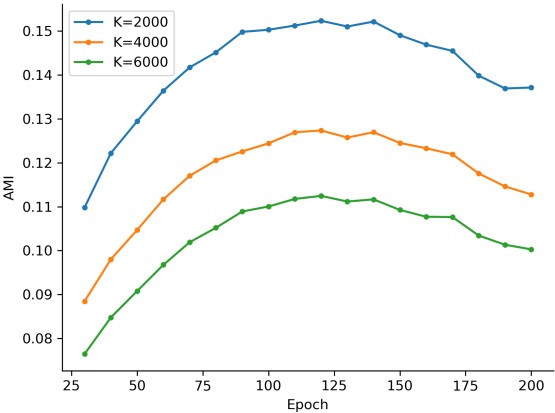

Figure 7: Adjusted mutual information between the clusters generated by Shape-PCL and ground-truth labels.

## E  TRAINING DETAILS FOR LINEAR CLASSIFIERS

**ImageNet-100**  We use the entire labeled training set of ImageNet-100 (100k images) to train the linear classifier on the fix representation. The linear classifier is trained for 100 epochs and the batch size is 256. The SGD optimizer is adopted, with the initial learning rate of 0.3 and momentum of 0.9. The learning rate will be multiplied by 0.1 at epoch 60 and 80.

**STL10**  We use the labeled training set of STL10 (8000 images) to train the linear classifier on the fix representation. The linear classifier is trained for 20 epochs and the batch size is 256. The SGD optimizer is adopted, with the initial learning rate of 0.3 and momentum of 0.9. The learning rate will be multiplied by 0.1 at epoch 12 and 16.

**imagenette**  We use the labeled training set of imagenette (10k images) to train the linear classifier on the fix representation. The linear classifier is trained for 20 epochs and the batch size is 256. The SGD optimizer is adopted, with the initial learning rate of 0.03 and momentum of 0.9. The learning rate will be multiplied by 0.1 at epoch 12 and 16.

