# OpenReview forum: "The benefits of Incorporating Shape Priors in Contrastive Learning"
_ICLR.cc/2024/Workshop/Re-Align — ICLR 2024 Workshop Re-Align Poster_

### Official Review · Reviewer_1qU6 · 2024-02-22
**A not entirely convincing hack**

**Rating:** 2
**Fit:** 2
**Confidence:** 2

**Workshop Review:**

This manuscript aims to bias contrastive learning DNNs towards using shape features by using an explicit estimate of the figure ground segmentation. Concretely, the authors extended prototype contrastive learning to use prototypes that are not based on the original images but based on shape masks that are passed through the momentum encoder corresponding to the network. The shape masks are generated by TRACER, another existing DNN. By itself the proposed method decreases performance on linear probe evaluations of the learned representations. It does seem to lead to faster training in the beginning though and does improve performance on style transferred images by a few %.

Overall, this manuscripts is around the border for me. The topic seems reasonably close to realigning human and machine perception as the authors are trying to improve shape bias, a well known point where humans and DNNs do not align currently. On the other hand, the shape masks passing through (moment encoder of) the same network seems decidedly not humanlike to me and more like a hack and the results for improving the shape bias are somewhat underwhelming. Especially, different contrastive learning objectives are needed to achieve good clean image performance. Additionally, the concrete results are a little thin, some of the details are lacking, most of the references are brocken, etc. Overall the presentation quality is not great.

**Reason For Not Giving Higher Score:**

The gains in shape bias are relatively small (only a few percent) and keeping high clean image performance requires standard contrastive learning approaches additionally. And there are actually quite few results in this paper, none of which are truly state of the art. And from the human side, I don't think this looks at all like a mechanism that might happen in human perception.

**Reason For Not Giving Lower Score:**

it is at least the right topic and there is some positive results here in terms of speed of training.

**Reviewer Domain:**

cognitive science

---

### Official Review · Reviewer_PXAg · 2024-02-26
**Interesting method to introduce shape bias, with some scalability limitations.**

**Rating:** 2
**Fit:** 3
**Confidence:** 2

**Workshop Review:**

The paper describes an application of contrastive learning to instill shape bias in neural networks. It utilizes clustering to create a shape prototype, which is then employed by a contrastive mechanism to enforce shape bias. This shape bias serves as a means to counteract the networks' natural tendency to focus on texture.

The paper provides an intriguing mechanism that is loosely based on how the human brain develops shape bias for object classification. I believe the paper offers a clear explanation of the method. The experiments seem to demonstrate that the method effectively imposes shape bias on the evaluated neural networks.

The main weakness is that scalability appears challenging; for larger datasets, generating such extensive bias masks may prove problematic. Perhaps a minor comment is that some of the tables would have been more effectively presented as graphs. Since they describe performance over epochs, a graphical representation might have been more illustrative.

**Reason For Not Giving Higher Score:**

The paper possesses significant potential and offers an elegant framework for introducing shape bias into neural networks. My main concern is that this method appears to necessitate providing a mask at training time, which could pose challenges when scaling up the experiments. Despite this, I believe that the workshop would greatly benefit from having this work presented, as it contributes valuable insights into the field.

**Reason For Not Giving Lower Score:**

I think the paper has a good demonstration of why shape bias is beneficial in training deep neural networks. It also provide some experiments that demonstrate the claims in different aspects.

**Reviewer Domain:**

cognitive science

---

### Decision · Program_Chairs · 2024-03-02

Accept (Poster)